# FusionScan: A Novel AI-Based Multi-Modal Imaging Technique for Enhanced Medical Diagnostics, Inspired by Stanford's Mini-Fellowship Program in Molecular Imaging

**Gurnoor Singh Dang**[1]                                GURNOORSD@GMAIL.COM
[1] *Student, The College Preparatory School, Oakland, CA 94618*
**Majid Rodgar**[2]                                       MAJRODG@STANFORD.EDU
[2] *Post-doctoral Scholar, Department of Genetics, Stanford School of Medicine, Stanford, CA 94305*
**Michael Snyder**[3]                                     MPSNYDER@STANFORD.EDU
[3] *Stanford W. Ascherman Professor of Genetics, Stanford School of Medicine, Stanford, CA 94305*

**Editors:** Accepted for publication at MIDL 2025

## Abstract

Modern medical diagnostics rely heavily on imaging technologies, each characterized by unique advantages and inherent limitations. Inspired by the insights from Stanford's Mini-Fellowship Program in Molecular Imaging Techniques, this research proposes FusionScan, an innovative technique that integrates Magnetic Resonance Imaging (MRI) and X-Ray Computed Tomography (CT) as imaging modalities along with an AI-based system. The research demonstrates how the fused system can provide unparalleled depth, resolution, and functional data by synthesizing the strengths of these two techniques, offering a robust solution for complex medical challenges. The goal is to fuse the images of the two modalities and analyze them using AI models, hence enhancing the resolution and specificity of the molecular images. Simulation results demonstrate the value of AI in enhancing medical images.

**Keywords:** Molecular Imaging, Magnetic Resonance Imaging (MRI), X-Ray Computed Tomography (CT), AI, Medical, Oncology, Diagnostics, Deep Learning, Fusion.

## 1. Introduction

Diagnostic imaging is critical in medicine but is often hampered by the limitations of existing technologies. Various existing imaging techniques offer distinct advantages, but they have specific limitations. PET and SPECT provide detailed molecular and metabolic data but suffer from moderate spatial resolution. Computed Tomography (CT) and X-ray deliver high-resolution anatomical structures rapidly; however, they expose patients to ionizing radiation and offer limited soft tissue contrast. Magnetic Resonance Imaging (MRI) excels in capturing soft tissue details without radiation but is costly to operate and requires longer scan times. FusionScan addresses this by offering an AI-enabled multi-dimensional imaging capability that no single technology currently offers, particularly in terms of integrating detailed molecular information with deep-tissue imaging.

## 2. Proposed Medical Imaging Technique: FusionScan

We are proposing a new technique called FusionScan, a blended design (MRI and CT) using AI to usher in a new era of molecular imaging by solving the resolution and depth issues. Figure 1 shows FusionScan's integrated hardware and software platform. Figure 2 shows the AI workflow and the feedback loop for iterative image reconstruction. Combining CT and MRI data with AI-driven enhancements, FusionScan can achieve unprecedented levels of detail and depth, far surpassing the capabilities of traditional modalities. FusionScan leverages advanced AI algorithms for image synthesis, utilizing techniques from generative AI to integrate and reconstruct complex data streams from multiple imaging modalities into coherent, high-definition images.

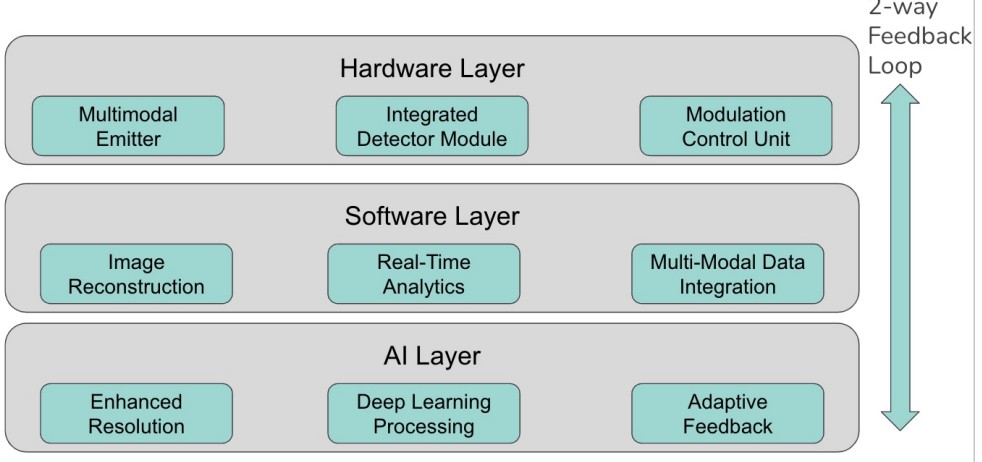

Figure 1: FusionScan System Architecture

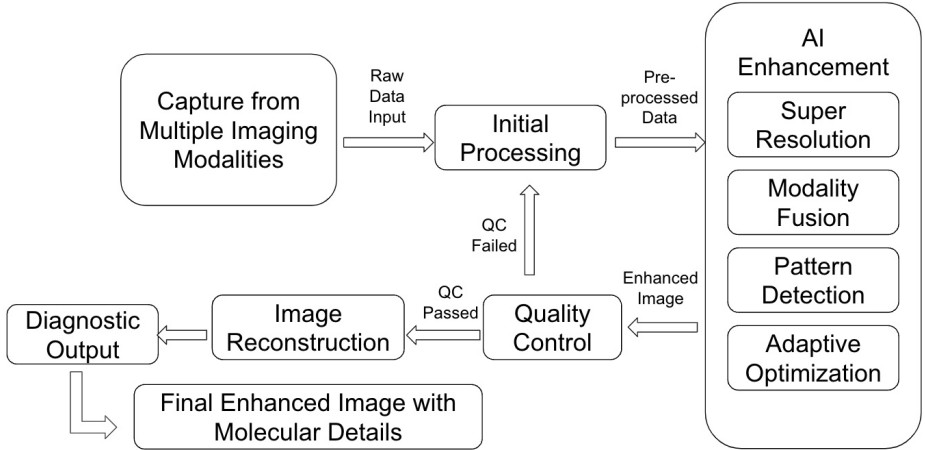

Figure 2: FusionScan AI Workflow

## 3. Simulation Experiments and Results

We conducted simulations to demonstrate the use of AI models, such as U-Net, GANs, and deep learning-based super-resolution models, to enhance medical imaging. Multiple MRI and CT scans were processed using this approach. Enhancement to brain MRI scans can show sharper contrast and clearer structural differentiation. T1-Weighted MRI with contrast after enhancement shows improved tumor visibility of the meningioma of the falx and differentiation of soft tissues (Figure 3). CT scan enhancement can show clearer soft tissue contrast and organ differentiation and improved organ visibility, aiding in rapid diagnosis. Non-contrast CT demonstrating multiple bilateral renal calculi (arrows) after enhancement shows improved stone detection and renal structural clarity (Figure 4).

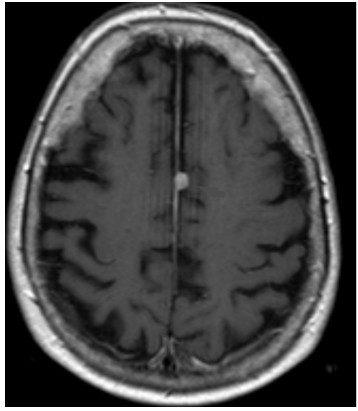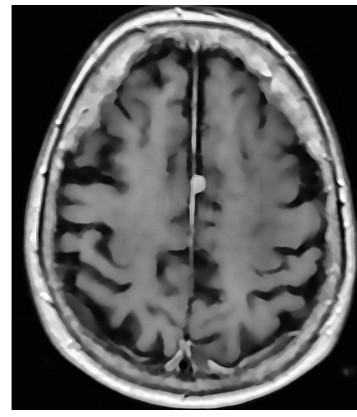

Figure 3: Brain MRI Scan With Tumor (Left: Original, Right: Enhanced)

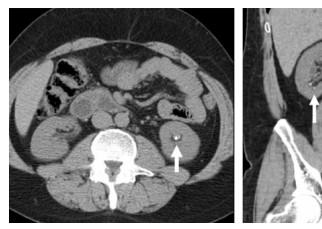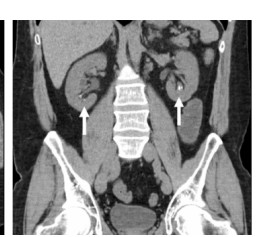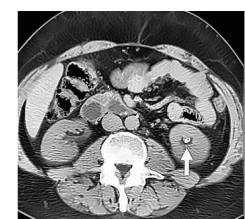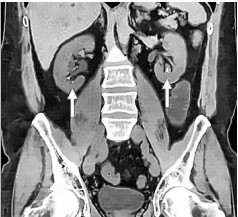

Figure 4: Non-contrast CT image of multiple bilateral renal calculi(arrows) (Left: Original, Right: Enhanced)

## 4. Conclusion

FusionScan represents a revolutionary step forward in medical imaging technology. By integrating multiple advanced imaging modalities and using AI models, it addresses key limitations in current diagnostics, offering a new horizon in patient care and medical research. This innovation, inspired by Stanford's Mini-Fellowship, exemplifies how advanced education and collaborative learning can lead to significant breakthroughs in medical technology.

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
