# OpenReview forum: "FusionScan: A Novel AI-Based Multi-Modal Imaging Technique for Enhanced Medical Diagnostics, Inspired by Stanford's Mini-Fellowship Program in Molecular Imaging"
_MIDL.io/2025/Short_Papers — MIDL 2025 - Short Papers_

### Official Review · Reviewer_xHey · 2025-04-25

**Rating:** 4
**Confidence:** 5

**Summary:**

FusionScan is a novel AI-driven imaging technique that integrates MRI and CT technologies to enhance medical diagnostics. Inspired by Stanford’s Mini-Fellowship in Molecular Imaging, it combines the strengths of both modalities with advanced AI models like U-Net and GANs to create sharper, more detailed medical images.

**Strengths:**

A major strength of FusionScan is its ability to synthesize detailed molecular and structural information, offering improved image clarity, resolution, and diagnostic accuracy. By leveraging AI, it reduces the individual limitations of MRI and CT while enhancing tissue differentiation and disease visibility.

**Weaknesses:**

One potential weakness of FusionScan is its complexity and cost; combining two imaging modalities and AI systems could make it expensive and technically challenging to implement widely. Additionally, reliance on AI models could introduce biases or errors if the algorithms are not rigorously validated across diverse patient populations.

---

### Decision · Program_Chairs · 2025-05-01

Accept